# A Pilot Study on the Feasibility and Usability of a Midwife-Led Visual Educational Tool for Sex Education

**DOI:** 10.3390/ijerph23010024

**Published:** 2025-12-23

**Authors:** Mayu Tokuoka, Hisao Nakai, Nobuki Shimaoka

**Affiliations:** Faculty of Nursing, University of Kochi, 2751-1 Ike, Kochi 781-8515, Japan; tokuoka_mayu@cc.u-kochi.ac.jp (M.T.); shimaoka@cc.u-kochi.ac.jp (N.S.)

**Keywords:** midwife, sex education, pilot study

## Abstract

Enhancing sexual literacy through sex education from an early age is vital for preventing unintended pregnancies and sexually transmitted infections. The “Beginning of Life” section of sex education presents a crucial opportunity to educate students on fertilization and fetal development. This study aimed to determine the feasibility and usability of the Beans Education Project Card (BEPC), a novel teaching tool for this section, in a real-world educational setting. Five midwives with sex education experience were provided with the BEPC for use with elementary, middle, and high school students over an approximate 3-month period (October 2024 to January 2025). Subsequently, group interviews were conducted with the midwives to gather feedback on the design, feasibility, and usability of the tool. The interview transcripts were analyzed using qualitative analysis, with large language model-assisted thematic exploration employed as a supplementary method to identify key themes. The analysis showed that the BEPC was highly regarded for its visual appeal, ease of use, and potential to engage families. The hierarchical design and visual representations, such as the use of small holes and beans to represent different stages of fetal development, were particularly effective in facilitating student understanding. The findings suggest that the BEPC has the potential to be a visually engaging tool for interactively teaching the process of fertilization and fetal development in the “Beginning of Life” part of sex education. Future research should focus on collaborations with parents and the development of strategies for reaching out to absent or home-schooled students.

## 1. Introduction

At the 1994 International Conference on Population and Development in Cairo, countries were called upon to address the educational and service needs of adolescents to enable them to manage their sexuality positively and responsibly. This conference prompted the promotion and implementation of youth-friendly sexual and reproductive health education [1]. However, 30 years after this pivotal meeting and despite substantial progress in society through people-centered development, several new risks pose challenges to sexual and reproductive health and rights [2]. One important example is the proliferation of sexual information facilitated by advances in information and communications technology, particularly through smartphones, the Internet, and social media platforms [3,4]. In the current digital landscape, young people increasingly tend to seek sexual information online, leaving them vulnerable to misinformation and, in some cases, severely harmful or inadequate content [5,6]. The sexual health literacy of young people is low and they are vulnerable to unintended pregnancies, abortions, and pregnancy-related complications, including sexually transmitted infections and AIDS [7]. There is compelling evidence that school-based sexuality education plays an essential role in safeguarding students’ sexual health and well-being, particularly regarding pregnancy and sexually transmitted infections [8]. Research suggests that visual and experiential foundational education targeting young people in school-based sex education can increase sexual literacy [9,10] and ultimately prevent unintended pregnancies, sexually transmitted infections related to unsafe sexual practices, and violence against young people.

Despite the promotion of high-quality and comprehensive sex education worldwide, as evidenced by the UNESCO 2009 publication and 2018 revision of the International Technical Guidance on Sexuality Education, substantial challenges remain [11]. The World Health Organization has noted that the global problem of adolescent pregnancy disproportionately affects less educated and poorer populations, leading to severe health, social, and economic consequences [12]. Teenage pregnancy can hinder a girl’s educational prospects and limit her economic opportunities [13]. Preventing teenage pregnancy has direct effects on the short-term and long-term well-being of both the individual and their family. Therefore, a broad range of initiatives involving families, schools, policymakers, and young people is needed for adequate prevention [14]. In one study, adolescents who received sexual and reproductive health education during adolescence were more likely to have sufficient knowledge to protect their sexual and reproductive health [15]. The study showed that as part of this education, collaboration with healthcare professionals who were well informed about sexual and reproductive health to provide more realistic and relevant education was effective as a supplement to education from school teachers [15]. Furthermore, educational methods should be interactive, allowing students to learn through fun and engaging experiences rather than simply providing knowledge through lectures. Such interactive learning is essential for motivating students to prevent pregnancy and engage in healthy sexual behaviors and to increase knowledge retention [16].

Although the adolescent abortion rate in Japan is lower than that of other developed countries [17], the number of induced abortions in Japan increased by 3.3% in 2023, with a particularly large increase among young people, especially those aged under 20 years, posing a major social problem [18]. Furthermore, the recent spread of sexually transmitted infections such as syphilis is a growing concern and requires urgent measures [19]. It is widely recognized that sex education for young people in Japan lags behind that of other countries [20]. Since the revision of the Course of Study in 2017, more comprehensive and up-to-date adolescent sex education from external lecturers such as obstetricians and midwives has been implemented in Japan [21,22]. Healthcare professionals can play a key role in promoting adolescent sexual health. By providing educational interventions, healthcare professionals can empower students to make informed decisions and reduce their risk of sexually transmitted infections and unintended pregnancies [23]. As part of sex education in Japanese schools, midwives primarily target elementary and junior high school students, and various creative approaches are used by individual midwives and school teachers to visually represent the sexual process from fertilization to fetal growth and birth [24,25]. In Japanese schools, the importance of sex education that involves collaboration between school nurses and highly specialized midwives is recognized [26]. Midwives are committed to providing education tailored to the individual needs of their students; however, standardized implementation methods and procedures have not yet been established, and these are currently left to the discretion of midwives and school teachers [27].

In this study, we focused on the “Beginning of Life” section of sex education, which covers fertilization and fetal development. Previous studies have demonstrated that learning about the human reproductive system using models and practicing correct condom use can increase adolescents’ knowledge of contraception, improve sexual health literacy, and positively affect future sexual behaviors [28]. Considering the effectiveness of experiential learning, we initiated the Beans Education Project with the aim of helping students to visualize the processes of fertilization and fetal development in the mother’s body and to facilitate students’ experiential learning and sharing, thereby helping to increase their sexual health literacy. There is evidence that early development of sexual health literacy can prevent unintended pregnancy [29] and risky sexual behaviors [30]. This project aims to promote sexual health literacy by providing students with an experiential learning opportunity to understand the beginning of life.

The objective of this study was to determine the feasibility and usability of the Beans Education Project Card (BEPC), a sex education tool used by midwives in the “Beginning of Life” section of sex education, by investigating its implementation by midwives engaged in sex education. There is a need to improve sexual literacy by including emotional literacy in comprehensive sexuality education in schools [31]. Sex education empowers students to make confident decisions about their bodies and sexuality [32]. The development of the BEPC could provide students with interactive and enjoyable learning opportunities, motivating them to engage in appropriate sexual behavior. Furthermore, the BEPC may help to standardize sex education implementation procedures, thus facilitating evaluation and reducing the workload involved in creating self-made teaching materials.

## 2. Materials and Methods

### 2.1. Study Design

A descriptive research design was used. Qualitative analysis was conducted on the data from group interviews, with topic modeling (large language model [LLM]-assisted thematic exploration https://gemini.google.com/app?hl=ja, accessed on 1 April 2025) employed as a supplementary method. The BEPC is shown in Figure 1, and the BEPC implementation procedure is shown in Figure 2.

### 2.2. Description of the BEPC

Sex education using the BEPC tool lasted approximately 20 min and was implemented as part of a broader comprehensive sexuality education program for students. This comprehensive program had a total duration of approximately 60 min and covered topics such as sexually transmitted infections, unintended pregnancy, interpersonal relationships, and gender-based issues. The content of the program was original and did not fully adhere to the UNESCO International Technical Guidance on Sexuality Education.

The BEPC consists of a paper card measuring approximately 10 cm by 15 cm. The card is designed to resemble a broad bean and features a small aperture on the left-hand side. A realistic illustration of an azuki bean (approximately 7.0–10.0 mm in length) is depicted on the right-hand side of the card to provide a visual reference for users. The BEPC allows students to visually and easily experience the growth process of a fetus by representing the size of a fertilized egg using a hole of approximately 0.1 mm in diameter, and by representing the growing fetus using illustrations of azuki beans and broad beans. Students can look through the hole to see how small a fertilized egg is. Subsequently, they can interactively learn about the beginning of life by looking at the illustration of the azuki bean and the shape of the card (which resembles a broad bean). After the lecture, students are allowed to take the BEPC home with them. This helps to promote communication between parents and children about the beginning of life and sexuality within the family, providing an opportunity for children and parents to discuss these issues in the home. The BEPC is designed so that instructors with different experiences and backgrounds can teach students about the beginning of life and the growth of the fetus in a standardized way. This could help to provide an educational approach to sex education that involves families, rather than being limited to the school environment. The BEPC is shown in Figure 1. It is important to note that this program and the BEPC tool are supplementary interventions focused on “The Beginning of Life” and are not intended to be a comprehensive curriculum covering all domains of the UNESCO International Technical Guidance on Sexuality Education (Table 1).

### 2.3. Procedures for Implementing Sex Education Using the BEPC

The following is a step-by-step guide for midwives to use the BEPC in teaching a “Beginning of Life” unit. A visual representation of how to use the BEPC is provided in Figure 2 along with a brief explanation.

Tell the students that you will provide a brief explanation of the beginning of life, including fertilization, and then distribute the BEPC to the students.Please dim the lights in the classroom. Then, ask the students to hold up the BEP to the fluorescent lights or the window and look through the holes in the cards.After the students have visually experienced the size of the holes, explain that the approximate size of the holes is 0.1 mm, which is roughly the size of a fertilized egg.After increasing the light in the room, please tell the students to shift their gaze to the illustration of the azuki bean next to the hole. Tell students that these beans are approximately 7.0 to 10.0 mm in length and represent a fetus at approximately 6 weeks of pregnancy. This is when a tiny heartbeat can first be detected.Next, discuss with the students the size of the BEPC itself, which is shaped like a broad bean. The card is approximately 15 cm long, and represents the crown-to-rump length of a fetus at approximately 19–20 weeks (4 months) of pregnancy. At this stage, the fetus has a distinctly human form, and many mothers begin to feel fetal movements.Finally, please ask the students to write their names on the back of the BEPC. Encourage them to take the cards home at the end of the day and discuss what they have learned with their families.

### 2.4. Data Collection

We conducted a snowball sampling study to recruit five midwives working in sexual education in elementary, junior high, and high schools in one prefecture in Japan. Participants were selected according to the following criteria: a minimum of 5 years of midwifery experience and at least 1 year of experience in sexual education. The purpose of this pilot study was to identify the range of diverse opinions about feasibility and usability. The interviews with the five experienced professionals provided sufficient information to fulfil this exploratory objective. The target prefecture has developed a publicly available protocol for sexual education, which is distributed to and followed by both physicians and midwives, who integrate their own expertise and knowledge with the protocol content [33]. Following the schools’ approval to incorporate the BEPC into their ongoing sexual education curriculum, midwives proceeded to implement the BEPC in the “Beginning of Life” section. The BEPC-implemented classes took place over an approximately 3-month period, from October 2024 to January 2025.

After the implementation, we conducted an online group interview using a semi-structured interview guide to evaluate the BEPC, identify areas for improvement, and assess its usability. Details of the interview questions are presented in Section 2.5. The online group interview was conducted using Zoom version 6.3.2 (Zoom Video Communications, San Jose, CA, USA) in January 2025.

### 2.5. Survey Items for Group Interviews

#### 2.5.1. Evaluation and Improvement of the BEPC

Impression of the BEPC (including their potential to capture the interest of students and their families): What do you think about the overall design and structure of the BEPC? Which aspects did you think were excellent, and which parts did you think were difficult to understand?

Feasibility of implementing the BEPC: How do you think the BEPC could be used in teaching the “Beginning of Life” section? In what specific situations do you think they would be useful? Compared with other teaching materials you have used, what are the advantages of the BEPC?

Challenges and areas for improvement: Are there any areas where you feel improvement is needed? Specifically, what aspects could be improved? What additional features or information would you like added to the BEPC?

#### 2.5.2. Usability

Ease of use: How easy was the BEPC to use? Were there any parts that were difficult to use? Was the tool intuitive?

Information quantity: Was the amount of information on the BEPC appropriate? Were there parts where you felt there was too much information or too little information?

Visual design: Was the design of the BEPC clear? Were the cards easy to use in lectures?

#### 2.5.3. Other Issues

Are there any other points you wish to make or opinions you have about the BEPC? Please feel free to share anything else you noticed.

#### 2.5.4. Analytic Methods

We used a hybrid analytical approach combining computational LLM-assisted thematic exploration with collaborative human interpretation to ensure the rigor and trustworthiness of our findings. The analysis proceeded in two main phases, outlined below.

Phase 1: Exploratory Theme Discovery. Initially, the verbatim transcripts were processed using the thematic exploration capabilities of an LLM (Google Gemini 1.5 via Google Chrome version 134.0. 6998.88, https://gemini.google.com/app?hl=ja, accessed on 1 April 2025). This served as an exploratory, inductive step to identify a broad range of potential latent themes within the data, minimizing initial researcher bias in theme identification.

Phase 2: Collaborative Validation and Refinement. Subsequently, the research team engaged in a rigorous deductive validation process. All authors collaboratively reviewed each theme generated by the model, comparing it against the original transcripts. Through an iterative process of discussion, themes were refined, merged, renamed, or discarded to ensure they were deeply and accurately grounded in the participants’ own words and narratives. The final thematic structure presented in this paper was established only after reaching a full consensus among the research team. As the research team was involved in the development of the BEPC, potential researcher bias was mitigated by ensuring that the analysis was conducted collaboratively by several authors and that the interpretations were thoroughly grounded in direct quotes from the participants. This hybrid methodology was chosen to leverage the computational power of LLM-assisted thematic exploration for comprehensive initial pattern detection while ensuring the final analysis was ultimately shaped by human-centered interpretation, contextual understanding, and collaborative scholarly judgment.

### 2.6. Ethical Considerations

This study was approved by the Institutional Review Board of The University of Kochi (No. 24–38).

All participating midwives provided written informed consent before the group interview. The educational sessions themselves were not a research intervention for the students but were conducted as part of the midwives’ professional practice within the existing school frameworks. As such, all safeguarding procedures were adhered to according to the established policies of the participating schools. These school-led procedures included protocols for parental notification, ensuring students’ right to decline participation (opt-out), and managing any student discomfort. The measures reported by the midwives, such as providing contact information or arranging follow-up with a school nurse, were implemented as part of this standard educational practice.

## 3. Results

### 3.1. Demographic Characteristics of the Participants

A total of five midwives participated in the implementation of the BEPC program. Four of the participants were staff midwives in different hospitals in one prefecture in Japan and one was a university faculty member specializing in maternal and child health nursing. The university faculty member had previously worked as a staff midwife in a hospital. All participants in the group interviews were women.

The age of the midwives ranged from 30 to 40 years; four were in their 40s and one was in her 30s. Their years of experience as midwives were as follows: 21 years, 16 years, 14 years, 11 years, and 7 years. Their experience in sex education was as follows: 15 years, 8 years, 6 years, 4 years, and 1 year.

The midwives implemented the BEPC program in the “Beginning of Life” section of the curriculum in various schools. One midwife taught 100 elementary school students at one school, whereas another taught 44 middle school students at one school. Two midwives taught a total of 153 junior high school students at two schools, and two midwives taught a total of 193 high school students at three schools (multiple responses). These schools were located in urban and suburban areas within the prefecture, representing a range of school sizes and student demographics. All schools were public.

### 3.2. Key Themes for Group Interviews

Regarding intervention fidelity, all five midwives reported during the group interviews that they had successfully followed the prescribed five-step procedure for using the BEPC. No major deviations from the core components of the intervention were reported by the participants. It should be noted that because this was a pilot study, a formal observational checklist to quantitatively measure adherence was not used. The assessment of fidelity was based solely on the midwives’ self-reports during the interviews. This methodological limitation is discussed further in Section 4.4. The qualitative analysis of the group interviews, supported by LLM-assisted thematic exploration, revealed that the BEPC is a highly effective visual tool for teaching reproductive processes in sex education. The analysis identified several key themes, including the BEPC’s high visual appeal, its ease of use, and its potential for home-based learning, which were consistently commended by the participating midwives. Furthermore, according to the midwives’ reports, students who received education using the BEPC showed signs of developing a more intuitive understanding of the reproductive process from fertilization to childbirth, suggesting a deepening of their knowledge about reproduction.

### 3.3. Evaluation of BEPC Design and Usability

#### 3.3.1. Visual Design and Appeal

The BEPC’s visual design was highly praised by participants for both its aesthetic appeal and its educational effectiveness. The bright, clean design, featuring a warm color palette that mimics the texture of a broad bean, was described as visually pleasing, welcoming, and approachable. One midwife described the tactile and emotional appeal by stating, “When I placed it in my palm, it really felt like a fetus, like a baby” (Midwife B). Another highlighted its biological resonance, noting, “The broad bean shape is nice; it resembles what you see on an ultrasound” (Midwife A). Furthermore, the visual hierarchy was identified as a key strength, with participants noting that the clear contrast between the 0.1 mm hole, the azuki bean illustration, and the overall card size provided an effective and intuitive representation of fetal development.

#### 3.3.2. Clarity and Information Design

The simplicity of the information design was a frequently praised feature. Participants reported that the amount of information was appropriate—neither excessive nor insufficient—which avoided information overload and facilitated focused learning. This simplicity was seen as a way to encourage deeper engagement, as one participant explained: “It’s simple, but there’s so much meaning packed inside. I think the simplicity actually encourages students to think more and imagine different things” (Midwife C). This focus on core concepts was further indicated by a suggestion to improve the tool’s clarity: “I think the Beans Education Project text would be better on the back. The shape itself can look like a baby, and I felt the text was a distraction” (Midwife D). The clear visuals and well-organized structure enabled students to rapidly access and comprehend the necessary information.

#### 3.3.3. Ease of Use

The physical attributes, such as its sturdy material, made the card durable and feel like something to be treasured. One midwife commented on this physical quality, stating, “The fact that the paper is thick is good. Unlike thin paper, it feels like something to be treasured and handled with care” (Midwife A). An important advantage noted by midwives was the logistical ease of use compared with other materials. As one midwife stated, “I think its biggest merit is the time it saves on preparation [compared with handmade materials]” (Midwife B). This was complemented by an intuitive layout, which participants noted had a shallow learning curve.

#### 3.3.4. Perceived Educational Effects

The theme of educational effects was characterized by four subthemes that described the educational effects of the BEPC: sparking interest, facilitating understanding, promoting communication, and applicability across age groups. The details of the analysis are shown below.

Sparking Interest: The unique design effectively piqued students’ curiosity and motivated them to learn. The card’s ability to engage even hard-to-reach students was highlighted by one midwife: “Even the second-year high school boys, the type who pretend to be asleep in class, were looking at the BEPC together with their friends” (Midwife D). Another described the shift in classroom atmosphere: “The students, who had been listening in silence, suddenly got a little excited and lively when they received the cards” (Midwife C).

Facilitating Understanding: By providing a visual representation of fetal development, the BEPC helped students grasp the abstract concept of size and scale. One participant explained the “gap” in understanding that the card bridges: “You can show a fertilized egg magnified on a PowerPoint slide, but I don’t think students can imagine its actual size. With this card, they see the real size, and I think they are surprised by that gap” (Midwife E). This was contrasted with passive learning, as another noted, “I saw them directly engaging with the object—holding it up to the light and really looking at it, which is different from just listening to a lecture” (Midwife A). Promoting Communication: Taking the BEPC home provided an opportunity for students to discuss the topic with their families. The card’s role as a social object was seen as a key benefit: “I think the best thing is that students can take it home and it becomes a usable conversation starter within the family” (Midwife A). This was echoed by another participant who observed, “Some students mentioned that they were going to show it to their parents” (Midwife B).

Applicability Across Age Groups: The results suggest that the BEPC could be an effective educational tool for a wide range of ages. One midwife was surprised by its effect on older students, noting, “I saw elementary school students having a lot of fun with it, but I was impressed that even middle and high school students were moved and looked at it with deep interest” (Midwife C). Another confirmed this broad appeal: “Even high school students have a genuine reaction of surprise, like, ‘Whoa, that’s amazing’” (Midwife D).

#### 3.3.5. Suggestions for Improvement

Two subthemes were identified as areas in which the BEPC could be improved: expansion of information and enhancing support for parents.

Expansion of Information: Participants suggested incorporating elements to make the learning experience more realistic, such as information about weight. One midwife described their current practice: “I supplement the card by also bringing a fetal doll to show them, “This is the actual size,” and “This is about the weight” (Midwife E). Another participant reinforced this idea, stating, “I thought it would be even easier for them to imagine the scale if information about weight was also included” (Midwife A).

Enhancing Support for Parents: The findings highlighted the need for strategies to engage parents. A direct suggestion was to involve them in the education itself: “What if we had parents come to the sex education class and experience this together with their children?” (Midwife B). The rationale for this was explained by another midwife: “The parents’ generation often hasn’t received much sex education themselves, so I think many of them would also be very surprised and learn things they never knew” (Midwife C).

## 4. Discussion

The results of this study suggest that the BEPC has the potential to be a highly feasible and usable tool for midwives teaching the “Beginning of Life” part of sex education. Below, we discuss (1) the design of the BEPC and its feasibility for use in sex education, (2) the usability of the BEPC, and (3) the challenges facing the implementation of the BEPC.

### 4.1. Design of the BEPC

Sex education materials often include illustrated books that explain physical anatomy and the beginning of life in an accessible way, as well as books designed to initiate conversations about sex between parents and children [34,35]. Technological advances have increased the range of materials that support visual and tactile learning about fetal development. For example, 3D/4D ultrasound imaging, which is widely used in obstetrics and gynecology, could be useful in sex education. However, access is limited to such technology, and it cannot be used in a family home setting. Recently, virtual reality representations of the experience of the fetus using information and communications technology [36] and smartphone applications have become available in sex education [37,38]. Despite the availability of these resources, a common limitation is their tendency to encourage passive receipt of information rather than active engagement. In contrast to these traditional approaches, the use of more hands-on, participatory tools that build upon the established effectiveness of interactive teaching methods in improving learning outcomes [39,40,41,42] may improve sex education.

Students’ perceptions of childbirth vary widely, and traditional lecture-style teaching may not adequately address this diversity. For instance, while some students may perceive childbirth as a mystical and miraculous event, believing that babies are gifts from heaven, research shows that others (even nursing students) believe in various myths about sex or have certain societal misconceptions, highlighting disparities in sexual literacy [43]. Some students may focus on the medical aspects of childbirth and experience fear or disgust at realistic depictions of it. Cultural norms and beliefs can also influence attitudes toward childbirth and sexuality, and gender differences may affect students’ understanding and perceptions [44]. Moreover, complex emotions surrounding sexuality, such as curiosity, fear, shame, and avoidance, are common among adolescents [45]. These diverse perspectives toward sexuality are likely shaped by a variety of factors, including personal values, religious beliefs [46], past experiences, and family environment. Given this complexity, the BEPC, a small handheld card distributed to each student, is particularly effective in accommodating the diversity of student perspectives on childbirth and sexuality by enabling personal reflection. Additionally, the card’s round and flexible shape may make it more approachable and inviting for children to use. The warm color scheme and appropriate information density can foster a safe space for students to explore the concept of childbirth.

According to the midwives who implemented the BEPC, its visual appeal was highly valued for increasing students’ understanding. Tomita reported that visually engaging handouts can increase students’ motivation to learn [47]. The BEPC allows students to visualize the gradual growth of a fetus from the size of a 0.1 mm diameter hole to the size of an azuki bean and then a broad bean. The card’s shape itself may have resonated with the image of a fetus, making the concept of birth more tangible. By actively participating in the lesson, students transformed the experience from merely watching and receiving information to one of touching and engaging, which may have helped to capture their interest more effectively. This could have facilitated a better understanding of the unseen phenomena occurring in the womb. Previous research has shown that positive childhood experiences can influence healthy behaviors, lifestyles, values, and actions in adulthood [48,49]. Building on this concept, it is hypothesized that an engaging, personal experience with the BEPC could contribute to the development of healthy attitudes. However, rather than speculating on long-term impacts, a more pragmatic next step is to evaluate the BEPC’s effect on measurable, intermediate outcomes. Therefore, future research should be designed to assess changes in students’ knowledge, self-efficacy (e.g., confidence in making healthy decisions), and behavioral intentions (e.g., the intention to communicate with parents about sexuality) immediately following the intervention. To this end, a more rigorous design, such as a cluster (school)-based trial, should be used, with measurements taken at a 4- to 12-week follow-up. A formal fidelity protocol must also be incorporated to ensure the intervention is delivered as intended. Such well-designed studies are essential for investigating the potential long-term effects of this educational tool.

The BEPC is designed to be taken home by students and discussed with their families. Students can write their names on the back of the card, and the midwives can encourage them to engage in conversations with their families after class. Research has shown that open communication about sexuality within families can deepen children’s understanding of sexual issues [50]. Parent-led sex education using simple materials has been shown to strengthen mother–child communication and is effective in preventing HIV [51]. By taking the BEPC home, students gain the opportunity to explain the card to their parents and share what they have learned, leading to more discussions about childbirth and sexuality within the family.

### 4.2. Usability of the BEPC

Generally, high usability is one of the most important factors for achieving a goal in human-centered solutions [52]. Research shows that educational materials that are highly usable can facilitate student understanding and improve learning efficiency [53]. A product is accepted by users only when its usability is at an acceptable level for the user [54]. The usability evaluation of the BEPC by midwives demonstrated its intuitive operation, shallow learning curve, clear information hierarchy, low error rate, and visually appealing design, which help to increase students’ motivation to learn.

In medical education, the difficulty of visualizing events related to embryology, such as fertilization and growth within the body, has long been recognized [55]. Advances in ultrasound imaging in medicine have enabled mothers and medical students to visualize these processes [56]. The fact that the BEPC, a simple card, can be used by both midwives and students to visualize fertilization and fetal growth is remarkable considering its simplicity and low cost. Midwives who provide sex education have a range of different experiences and backgrounds. Therefore, tools like the BEPC that are intuitive to use and contain an appropriate level of information are very effective in helping midwives of all backgrounds and experience levels to explain the fetal development process to students in an easy-to-understand manner. In particular, it is expected that the BEPC could contribute to deepening students’ understanding by helping them to visualize fetal growth in the womb.

### 4.3. Challenges Facing the Implementation of the BEPC

Although the BEPC effectively helps students to visualize fetal development from the point of fertilization, the midwives participating in the present study identified an important limitation: the BEPC does not provide tactile experiences of a baby’s weight or the actual morphological changes that occur in fetal development. To address this limitation, we recommend that interactive education using the BEPC should be combined with hands-on experiences using traditional fetal models. By learning about a baby’s actual shape and weight and receiving detailed information about fetal characteristics and maternal changes at each developmental stage, learners can obtain a more realistic understanding of the complex process of birth. Previous studies have demonstrated the effectiveness of using various teaching materials in reproductive and pregnancy education [57]. Therefore, there is a need to explore the development of educational methods that combine the visual effects and imagination-stimulating capabilities of the BEPC with fetal models.

This evaluation of the implementation of the BEPC by midwives highlights the importance of providing workshops and information to parents to facilitate communication with their children about sexuality. To prevent teenage pregnancy, open communication between parents and children about sexuality is needed, but there is evidence that both daughters and mothers experience difficulties in communicating about sex [58]. Considering the positive effect of in-home communication about sexuality on children [50], expanding the BEPC to include parents is an important task. Parents who are familiar with the BEPC may be motivated to use the tool for in-home communication about sex and pregnancy. This is an important area for future research. Although this study suggests that the BEPC can be adapted for use with students of a wide age range (from elementary to high school), it is likely that students’ understanding and perception of sex education vary according to age. Therefore, additional detailed research is needed, including specific considerations of how the BEPC could be used in a range of teaching methods to inform students about the different fetal developmental stages.

### 4.4. Limitations of the Study

This pilot study has several limitations. First, there were substantial limitations regarding the sampling method and context. Recruitment of only five midwives via snowball sampling from a single prefecture results in a high risk of selection bias and low generalizability of the findings. Although we have added a general description of the school settings (e.g., urban and suburban), we deliberately avoided providing more specific details about each school (e.g., exact size, specific socioeconomic indicators) to protect the anonymity of participating institutions and students, which is a critical ethical consideration in a study of this nature. Furthermore, regarding transferability, the small sample size (N = 5) precluded any meaningful analysis of differences by educational level or midwife experience. Although no such patterns emerged from the data, such differences are an important area for future investigation with a larger, more diverse sample. Therefore, the results should be interpreted not as generalizable evidence but as foundational insights from a feasibility and usability study, which will inform the design of a more rigorous, larger-scale trial. Another important limitation is the lack of a formal, quantitative fidelity assessment. Second, although the midwives reported adherence to the protocol in post-implementation interviews, we did not use methods such as direct observation or checklists to formally measure and quantify adherence (e.g., as a percentage). Therefore, we cannot definitively confirm the degree to which the intervention was delivered as intended. Incorporating a rigorous fidelity protocol will be a crucial component in the design of a future, larger-scale evaluation. Third, a further limitation of this study is its reliance on qualitative data to assess feasibility and usability. The study design did not include objective, quantitative indicators such as process metrics (e.g., actual session duration, unit costs) or a standardized usability scale (e.g., the System Usability Scale). The absence of these metrics means that our assessment of feasibility and usability is based on subjective interpretation rather than objective measurement. Future, larger-scale evaluation studies should incorporate these quantitative metrics to provide a more multifaceted assessment. As a template for such research, Appendix provides a list of recommended process and implementation metrics that should be collected. Fourth, the lack of uniformity among the included schools in terms of grade levels, sizes, and socioeconomic backgrounds may have introduced selection bias and limited the generalizability of the findings. Fifth, since the BEPC was developed by the research team, researcher bias may have affected its design and implementation. The use of snowball sampling may also have limited the sample’s representativeness. Sixth, while LLM-assisted thematic exploration was used as a supplementary tool to aid the primary qualitative analysis, this automated technique has inherent limitations. Natural language processing may not fully capture the contextual subtleties and emotional nuances of human language. Although the authors collaboratively reviewed and refined the themes to mitigate this, the initial automated step might have overlooked some aspects of the midwives’ feedback, potentially influencing the subsequent qualitative interpretation. Seventh, a further limitation of the qualitative process was the lack of member checking, as the final themes were not presented to the participants for validation, which could have enhanced the credibility of the interpretations. Eighth, this study focused on an initial version of the BEPC in one region of Japan, which restricts the generalizability of the findings to other cultural contexts. Ninth, further research is needed to assess the BEPC’s effectiveness as a preventive education tool, particularly concerning its impact on discussions about sex and its appropriateness for specific age groups. Finally, as a qualitative descriptive study, the findings provide in-depth insights but are not intended to be generalizable to the broader population of midwives or other educational contexts.

## 5. Conclusions

This pilot study identified four key findings. First, the study provides preliminary evidence suggesting that the BEPC, a novel sex education tool designed by midwives, shows substantial promise for increasing students’ understanding of sexuality and facilitating family communication. Second, the visual representation used in the BEPC, particularly the gradual progression of fetal growth depicted on the card, effectively captured students’ attention and facilitated a deeper understanding of the complex process of fertilization and fetal development. This visual approach was found to be more engaging than traditional lecture-based methods, particularly for students who may struggle to grasp abstract concepts. Third, the BEPC’s design focuses on individual reflection and engagement and effectively accommodates the diverse perspectives and emotional responses that students often bring to sex education. By allowing for personal exploration and reflection, the BEPC created a safe and inclusive learning environment for students to tackle their thoughts and feelings about sexuality. Finally, a unique feature of the BEPC is that students are permitted to take the card home to discuss it with their families. This feature could have a substantial positive effect on family communication about sexuality. By encouraging open dialogue between parents and children, the BEPC could contribute to creating a more informed and supportive family environment surrounding sexual health.

Implications of this research include the following. The positive initial evaluation of this tool suggests that the BEPC is a potentially useful tool for widespread implementation in school-based sex education programs across various student age groups. Future research should explore expanding its scope to include parents and caregivers, and developing workshops and resources to guide them in using the BEPC to facilitate family discussions about sexuality. Furthermore, the BEPC could be integrated into preconception care programs to educate future parents about pregnancy and childbirth. Addressing the limitations of this study, such as increasing the sample size and recruiting more diverse samples, is essential to further validate the effectiveness and generalizability of the BEPC across diverse educational contexts. Although this study suggests the potential of the BEPC as an educational tool, its widespread adoption requires further consideration. A critical next step, following a more rigorous evaluation of the tool’s effectiveness, will be a detailed analysis of production costs, logistics, and scalability models, such as offering a downloadable version or an open-license template.

In conclusion, although additional research is needed to fully evaluate its long-term effects, the BEPC demonstrated considerable potential as a valuable tool for enhancing sex education. By combining engaging visuals with interactive elements and encouraging family communication, the BEPC could contribute to a more comprehensive and effective approach to sex education that empowers students with knowledge, fosters open communication, and promotes healthy sexual development.

## Figures and Tables

**Figure 1 ijerph-23-00024-f001:**
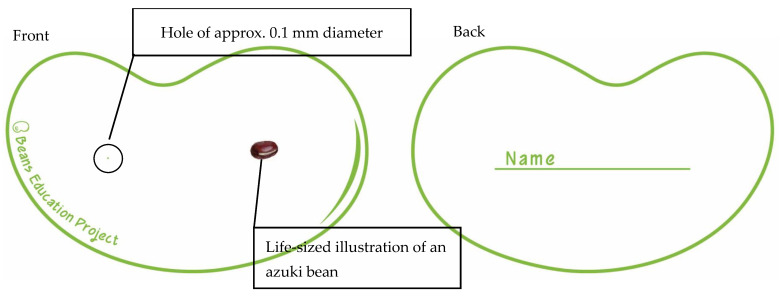
The Beans Education Project Card (BEPC). The card, shaped like a broad bean, measures approximately 10 cm by 15 cm. It is printed on a sturdy, matte card stock suitable for handling by students. It features a 0.1 mm diameter hole on the left (to represent the size of a fertilized egg) and a life-sized illustration of an azuki bean (approx. 7.0–10.0 mm) on the right (to represent a fetus at 6 weeks).

**Figure 2 ijerph-23-00024-f002:**
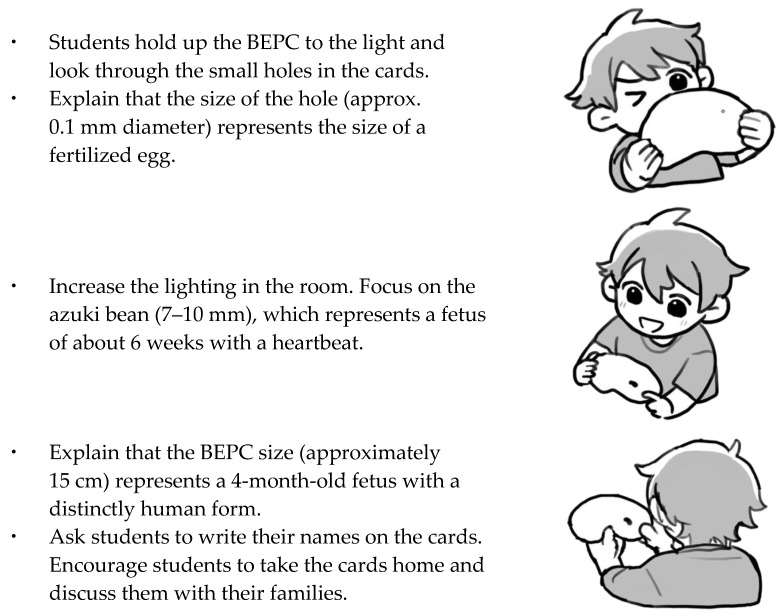
A concise guide and visual representation of BEPC usage.

**Table 1 ijerph-23-00024-t001:** Scope of the Beans Education Project Card (BEPC) Tool.

Category	Description
What BEPC Covers	The biological process of fertilization and fetal development (The Beginning of Life).
What BEPC Does Not Cover	Broader sexuality education topics, such as contraception, consent, gender identity, or interpersonal relationships.
Rationale	The BEPC is designed to be a supplementary, single-topic tool, not a comprehensive curriculum. Its purpose is to increase understanding of one specific biological process through a tangible experience.

## Data Availability

The interview transcripts generated and analyzed during the current study are not publicly available to protect participant privacy. The BEPC and its associated materials (e.g., step-by-step guide, fidelity checklist) are currently in a prototype stage and have not been widely distributed. However, requests for access to these materials from bona fide researchers or practitioners for collaborative purposes will be considered on a case-by-case basis, following a direct inquiry to the corresponding author.

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
