# Peer review of "A Pilot Study on the Feasibility and Usability of a Midwife-Led Visual Educational Tool for Sex Education"

_ijerph, 2025, doi:10.3390/ijerph23010024_

Round 1
Reviewer 1 Report
Comments and Suggestions for Authors
- This paper was well presented and well written (I am impressed with Google Translate!).
- My only reservation is whether the authors were able to draw any contrasts between the acceptability of the method across the different grade levels. Did it seem too simplistic for high school students or too complex for the elementary school students? If there were contrasts made, it would be helpful to add to the paper.
Author Response
Response to Reviewer 1
We are grateful for the time and effort that you dedicated to providing your valuable feedback on our manuscript. We have carefully considered all your comments and revised the manuscript accordingly.
- This paper was well presented and well written (I am impressed with Google Translate!).
Thank you for your kind words regarding the writing and presentation. Our process involved using Google Translate as an initial aid, followed by a thorough proofreading and editing process conducted by a native English speaker to ensure accuracy and clarity.
- My only reservation is whether the authors were able to draw any contrasts between the acceptability of the method across the different grade levels. Did it seem too simplistic for high school students or too complex for the elementary school students? If there were contrasts made, it would be helpful to add to the paper.
Thank you for this very insightful question regarding the tool's acceptability across different grade levels. This is a critical point, and we reviewed our interview data specifically with this question in mind.
Interestingly, the feedback from the midwives was contrary to this reservation. We found no evidence in the interviews to suggest that the tool was perceived as being too simplistic for high school students or too complex for elementary school students. On the contrary, the midwives reported that the core teaching mechanism—the tangible and visual process of moving from the tiny hole to the azuki bean and the broad bean shape—was universally effective. They felt this approach allowed the abstract concept of “The Beginning of Life” to be conveyed in a real and tangible way that resonated with students regardless of their age. This finding is supported by quotes in our Results section, such as one midwife noting her surprise that "even middle and high school students were moved and looked at it with deep interest."
Based on this positive feedback about its wide applicability, we are now considering expanding the tool's use to even younger age groups, such as preschoolers, in future iterations. Therefore, because no substantial contrasts emerged from the data, none have been added to the paper; instead, the tool's broad applicability is highlighted as a key strength.
Reviewer 2 Report
Comments and Suggestions for Authors

Author Response
Response to Reviewer 2
We are grateful for the time and effort that you dedicated to providing your valuable feedback on our manuscript. We have carefully considered all your comments and revised the manuscript accordingly.
Major Comments
- Study Design and Sampling. Only five midwives, recruited by snowball sampling, from a single prefecture → high risk of selection bias and low contextual variability. Suggest justifying the sample size (theoretical saturation does not apply to a single group; explain the sufficiency criterion) and add a description of the centers/schools (size, socioeconomic level, urban/rural).
Thank you for your insightful comments. We agree with your assessment. As you correctly pointed out, this was a pilot study (brief report), and we acknowledge that selection bias stemming from the sampling method and lack of contextual diversity are important limitations to the validity of our findings.
To address this, we have first added a general description of the school settings in the Results section (Lines 276–277), to the extent possible, while ensuring the anonymity of participating institutions. Furthermore, to address the fundamental limitations you highlighted, we have substantially revised and expanded the Limitations section (Lines 471–513). In this section, we now elaborate on the constraints of the sample size and sampling method, as well as the ethical considerations that led us to deliberately avoid providing specific background details to protect participant and institutional anonymity.
We believe these revisions help to clarify both the implications of our findings and the necessary caution in their interpretation. The revisions in the manuscript have been highlighted in red for ease of review.
- Intervention Description (BEPC) and Fidelity. Good detail on materials and the 5-step procedure, but a fidelity (adherence) section is missing: Did all midwives follow the script exactly? Were there any variations? I propose adding a table of essential components, materials, and allowable deviations, and reporting adherence (%) per session.
Thank you for this critical point regarding intervention fidelity. You are correct that we did not include a formal, quantitative fidelity assessment in the design of this pilot study. Because our primary goal was to evaluate the initial feasibility and usability of the BEPC, a formal adherence protocol with checklists or observation was beyond the scope in this preliminary phase. However, we recognize the importance of this issue. Based on your suggestion, we have taken the following two actions in the revised manuscript:
- We have added a statement to the Results section clarifying that, based on the qualitative data from our post-implementation group interviews, all midwives reported having followed the five-step procedure without substantial deviations (Lines 279–282).
- We have explicitly added the lack of a formal fidelity assessment as a key limitation in the Discussion section. We state that future, larger-scale studies will need to incorporate a rigorous fidelity protocol, as you suggested (Lines 471–513).
- Qualitative Methods: Reporting Standards. The study is presented as "descriptive" with group interviews + theme modeling. It lacks explicit mapping to the COREQ (or SRQR) checklist: researcher roles, prior relationship with participants, interview guide, duration, context, coding procedure (inductive/deductive?), discrepancy handling, use of software, supporting quotes, and evidence of member checking. I recommend adding an appendix with a guide, analytical flowchart, and "theme-subtheme-quote" table.
Thank you for the detailed feedback on our qualitative reporting. As recommended, we have revised the manuscript to better align with COREQ guidelines. This includes expanding the Methods section to clarify our analytical process, adding the interview guide and a theme-quote matrix as appendices, and noting the lack of member checking as a limitation.
- Topic Modeling Integration. Gemini 1.5 Flash is used for modeling as a support, but the analytical integration remains vague (how it was parameterized, number of topics, optimal selection, examples of top words, how it influenced human coding). I propose: (a) moving it to the Methodological Appendix with hyperparameters, k criterion, topic consistency, word clouds, and (b) clarifying that human qualitative analysis takes precedence and how disagreements were resolved. Add an audit trail.
Thank you for prompting us to clarify our analytical process. We have substantially revised the “Analytic methods” section to provide full transparency, as you have suggested.
We have now clearly articulated that we used a hybrid approach. Topic modeling (using Gemini 1.5) was used as an initial, inductive step for exploratory theme discovery. This computational phase was followed by a rigorous, deductive process where all authors collaboratively validated, refined, and reached a consensus on the final themes to ensure they were firmly grounded in the participant data. We have also explicitly described how discrepancies were handled through this consensus-based discussion.
We believe this detailed description now clarifies the distinct and complementary roles of the computational tool and our research team's interpretive analysis, addressing your concerns about the analytical procedure. However, providing specific topic modeling parameters (e.g., k, coherence score, word lists) does not apply to our methodology. We used a large language model (Gemini 1.5) for exploratory discovery, not a traditional statistical model like Latent Dirichlet Allocation, where such parameters are set and reported. The rigor of our method comes from the extensive human validation phase, as now described in the manuscript.
- Results: Evidence and Transfer. The findings are mostly summarized statements without illustrative quotes. Include ≥2 short quotes per subtopic to support inferences (aligned with COREQ). Report differences by educational level (primary/middle/high school) and by midwife experience, if they emerged.
Thank you for this feedback on strengthening the evidence in our Results section. We have addressed your two points as follows:
- Illustrative Quotes: As you suggested, we have now woven short, illustrative quotes directly into the main text of the Results section (3.3) to support the interpretation for each subtheme. A more comprehensive matrix of all themes and supporting quotes can be found in the new Appendix B.
- Subgroup Differences: We also re-examined the data for potential differences based on educational level or midwife experience. However, given the very small sample size (N=5), we did not identify any systematic patterns that could be reported without a large risk of over-interpretation. We agree that this is an important question and have highlighted it as a direction for future research using a larger sample.
- Minimum quantifiable outcomes (feasibility/usability). As this is a "feasibility and usability" study, a minimum set of objective indicators is missing: number of sessions, class size, actual duration, materials per student, incidents/adaptations, unit costs, losses, and a usability metric (e.g., SUS adapted to the faculty), even if exploratory. Propose a table of "process/implementation metrics" and, if none exist, declare this as a limitation and plan for the next study.
Thank you for highlighting the absence of quantitative feasibility and usability metrics. We agree that this is a key limitation of our pilot study. Accordingly, we have now added a new paragraph to the “Limitations” section (Lines 497–503) to explicitly address this point and to state that these metrics will be incorporated in future studies.
- Curriculum Alignment and Sensitive Content. The manuscript itself admits that the program "does not fully adhere" to the UNESCO guidelines; it is important to clearly specify which domains it covers and which it does not, and justify why (e.g., identity, consent, diversity, contraception). Include a BEPC ↔ UNESCO domain mapping as an annex.
Thank you for raising this important point about curriculum alignment with the UNESCO guidelines. We agree that situating our tool within this framework is an important consideration.
However, we must clarify that the BEPC is not designed to be a comprehensive, standalone sex education curriculum. Rather, it is a supplementary teaching tool designed to be used for one specific topic within a broader program: “The Beginning of Life.” Therefore, creating a direct mapping table against the comprehensive UNESCO domains, which cover a vast range of topics including consent, diversity, and contraception, would be methodologically inappropriate and potentially misleading. Such a table would inaccurately frame the BEPC as a poor curriculum by highlighting the many domains it does not cover, when its explicit purpose is to be an effective tool for a single, narrow topic. The BEPC does not cover these other domains simply because that is not its intended pedagogical function.
To address the spirit of your comment, we have now added a sentence in the Methods section 2.2 to more explicitly state the BEPC's narrow focus and to clarify that it is not intended to align with the entire UNESCO framework.
- AI Use and Transparency Declaration. It's correct to declare Google Translate and Gemini; however, for best practices, add: quality control (human double-checking), data leakage (do not enter sensitive data into cloud services), and model version/date.2
Use of Artificial Intelligence (Lines 573–583)
Thank you for the suggestions on best practices for reporting AI use. We have revised our “Use of Artificial Intelligence” declaration to be more explicit, as you recommended. Specifically, we have now: (1) made our description of the human quality control process more direct; (2) added a statement regarding our data anonymization process to address data privacy concerns; and (3) included the month of access for the AI model version.
- Discussion: Avoiding Overstatements. Some inferences about future impact on parental behaviors/adult attitudes are speculative; I suggest qualifying them and linking them to a pragmatic curriculum (cluster-school) with intermediate outcome measures (knowledge, self-efficacy, intention) and follow-up.
Thank you for your valuable suggestion to avoid overstatement in the Discussion. We have revised the relevant paragraph (in section 4.1) accordingly. The speculative claims about long-term impacts have been removed. Instead, as you recommended, we now frame this as a hypothesis for future research and explicitly state that the pragmatic next step is to measure intermediate outcomes such as self-efficacy and behavioral intentions.
Minor comments
- Figures 1–2: Incorporate more descriptive captions (exact dimensions, materials, legible photos/diagrams) and ensure the 0.1 mm hole with scale is visible.
Thank you for the feedback on improving our figures. We have revised Figure 1 and its caption to be clearer and more self-contained. (1) The caption now includes the exact dimensions and the purpose of each component, as you suggested. (2) To best illustrate these details, we have opted to use a labeled diagram rather than a photograph, as it more clearly represents the design features. We believe these combined changes make the figure much more informative.
- Terminology: BEPC/BEP are used interchangeably in some places; standardize acronym.
Thank you for pointing out this inconsistency. We have now standardized the acronym and use 'BEPC' consistently throughout the manuscript.
- Editing: Prune repetitions in 3.3 and 3.4; consolidate redundant subtopics (e.g., “visual appeal” appears in both design and usability).
Thank you for the feedback. Following your advice, we have consolidated the redundant subtopics from sections 3.3 and 3.4 into a single, streamlined Results section to eliminate repetition.
Specific suggestions for improvement
Add Annexes:
- Complete interview guide + theme-subtheme-quote matrix.
As you recommended, we have now added two new appendices to the manuscript. Appendix A contains the complete interview guide, and Appendix B provides a “theme-subtheme-quote matrix,” as you suggested.
- Analytical protocol: flowchart from transcription → preprocessing → coding → validation; topic modeling parameters (k, coherence metric, top-10 word list per theme).
Regarding the analytical protocol, the revised “Analytic methods” section now details our two-phase hybrid process, which serves as a textual description of the workflow. However, providing specific topic modeling parameters (e.g., k, coherence score, word lists) does not apply to our methodology. We used a large language model (Gemini 1.5) for exploratory discovery, not a traditional statistical model such as Latent Dirichlet Allocation, where such parameters are set and reported. The rigor of our method comes from the extensive human validation phase, as now described in the manuscript.
- BEPC ↔ UNESCO mapping table (what the card covers and what it doesn't).
Whereas we appreciate this interesting suggestion, a detailed mapping of our single tool against the comprehensive UNESCO guidelines is beyond the scope of this preliminary feasibility study. We have clarified in the manuscript that the BEPC is intended as a supplementary tool for one specific topic (“Beginning of Life”), not as a comprehensive curriculum intended to cover all aspects of the UNESCO framework.
- Implementation table (new in Results): Number of sessions per educational level, class size, actual duration, % adherence to the 5 steps, incidents, consumables per student, estimated cost per unit.
As stated in our response to comment 6 and our newly added limitation, these specific quantitative process metrics were not collected during this pilot phase. We have explicitly identified this as a key limitation and a requirement for future research.
We hope this clarifies how we have addressed your suggestions by either implementing them or explaining the methodological and scope-related reasons why certain items do not apply to this specific study.
- Illustrative Quotes. For each key subtheme, provide 1–2 (anonymized) quotes from midwives to support claims of “high appeal,” “ease,” “cost barrier,” and “home-based learning.”
Thank you for the feedback. We have now woven illustrative quotes into the Results section to provide direct evidence for our findings, as requested.
- Safeguarding Section. Procedures for addressing discomfort, sensitive questions, parental objections, and the inclusion of absent or homeschooled students (the manuscript itself suggests this in future lines).
Based on your feedback, we have added a “Safeguarding and Ethical Procedures” section to the Methods with a detailed explanation of our protocols.
- Clarify cost and scalability. Since cost is reported as a barrier, estimate the cost per card (materials/printing), shelf life, and logistics; consider a downloadable version/open-license template for printing in schools.
Thank you for raising these critical points regarding cost and scalability. We agree that these are crucial factors for real-world implementation and widespread adoption of the BEPC. However, the primary aim of this initial pilot study was to assess the feasibility and usability of the tool from the midwives' perspective, before a more formal evaluation. Therefore, a formal cost or scalability analysis was beyond the scope of this preliminary work.
Specifically addressing your points:
- Cost Estimation: We did not collect data on unit costs or logistics. Any cost estimate based on our small-scale prototype production would not be a meaningful or accurate reflection of potential costs at scale and could be misleading.
- Scalability and Licensing: Considerations such as offering a downloadable version or an open-license template are excellent ideas for future dissemination. However, we believe such decisions on a distribution model should follow a more rigorous evaluation of the tool's educational effectiveness.
We have now explicitly mentioned at the end of the Discussion section that a thorough analysis of cost, logistics, and scalability models is a critical next step for future research, once the tool's efficacy has been more formally established.
Round 2
Reviewer 2 Report
Comments and Suggestions for Authors

< !--a=1-->< !--a=1-->
Author Response
Response to Reviewer 2
Thank you for your thoughtful and detailed comments, and for the opportunity to revise our manuscript. We have addressed all feedback and believe the revised manuscript is significantly improved. Our detailed responses are provided below.
Study design and sampling
What has been done: You acknowledge the pilot nature, small N, snowball sampling, and single-
context setting; you expanded the Limitations and added a brief aggregated description of the schools.
What is still needed: Add 2–3 sentences in Methods justifying operational sample sufficiency for a
pilot (not aiming for saturation). Include 1–2 additional non-identifying contextual features (e.g.,
approximate school size range, public/private). Make the potential developer bias explicit and how
it was mitigated (collaborative analysis, use of quotes).
As per your suggestion, we have added sentences in the "Methods" and "Results" sections. The additions and changes are highlighted in red font in the revised manuscript.
Intervention and fidelity (adherence)
What has been done: The five-step procedure is clearly described, and adherence is reported via midwives’ self-report.
What is still needed: Add an appendix table of essential components and adherence (component, minimum materials, non-negotiables, allowable variations, compliance indicator). In Methods, state explicitly that there was no observational checklist given the pilot phase, and cross-reference the Limitations.
Following your suggestion, we have added Appendix C to the manuscript. Furthermore, we have added text to the Results section.
Qualitative methods and COREQ/SRQR standards
What has been done: You expanded the analysis section, added the interview guide and a theme–subtheme–quote matrix in appendices, and acknowledged the absence of member checking.
What is still needed: Provide a brief COREQ map (table or appendix) summarizing: team roles and any prior relationship with participants, setting/duration, coding approach, how discrepancies were resolved, software (if any), and justification for no member checking. Report average interview duration and corpus size (words/pages), and note how consensus was documented (e.g., memos).
In response to your comment, we have added a summary table for the key items of COREQ as Appendix D.
LLM-assisted thematic exploration (formerly “topic modeling”)
What has been done: The hybrid approach is clearly explained and the LLM’s exploratory role is distinguished from LDA (no k/coherence required).
What is still needed: Rename the label to “LLM-assisted thematic exploration” to avoid confusion.
Add an audit trail (generic prompt, a non-sensitive sample output, and its disposition:
accepted/modified/rejected). Align the model access date with the analysis period.
Following your guidance, we have replaced the terminology, renaming the section to 'LLM-assisted thematic exploration' throughout the manuscript. We have also included Appendix E to provide the requested audit trail, which covers the generic prompt, a sample output, and the disposition of the AI-generated themes.
Results: evidence and transferability
What has been done: Short illustrative quotes have been integrated into the text, with a
comprehensive matrix in the appendices.
What is still needed: Verify that all subcategories have ≥2 quotes (in-text or in the matrix). Add a transferability paragraph stating that, with N=5, no robust patterns by educational level or midwife experience were observed, to prevent over-interpretation.2
We have reviewed Appendix B and the main text, and, as per your suggestion, we have added the necessary sentences to the relevant sections.
Minimum feasibility/usability metrics
What has been done: You acknowledge as a limitation the lack of objective indicators at this stage.
What is still needed: Include (even as a template for future studies) a process metrics table: number of sessions by educational level, class size, actual duration, incidents/adaptations, consumables per student, estimated unit cost, and adherence to the five steps (self-report).
We have added Appendix F, which includes the process metrics template, and referenced it in the Discussion section.
Curricular alignment and sensitive content (UNESCO)
What has been done: You clarify that BEPC is a supplementary tool focused on “Beginning of Life,”
not a comprehensive curriculum.
What is still needed: Add a declarative scope box (what BEPC covers and does not cover, and why).
This addresses the reviewer’s request without implying alignment with all UNESCO domains.
We have added a Table 1 in the Methods section to clearly delineate what the BEPC covers, what it does not, and the rationale for its narrow focus, as you suggested.
AI use and transparency
What has been done: A more explicit statement on human quality control, anonymization, and the
model’s temporal reference.
What is still needed: Present it as a checklist: (i) anonymization prior to any external service, (ii)
double human verification, (iii) model name and month/year of access, (iv) limited scope of AI
(exploratory support; primacy of human analysis). Check date consistency.
We have revised the "Use of Artificial Intelligence" section as per your suggestion.
Discussion: prudence and roadmap
What has been done: Claims have been toned down and intermediate outcomes are proposed.
What is still needed: Add two sentences to specify a roadmap: cluster (school)-based design,
measures (knowledge, self-efficacy, intention), follow-up window (4–12 weeks), and inclusion of a
fidelity protocol.
We have incorporated the suggested specific details—covering the proposed design, measures, follow-up window, and fidelity protocol—into the roadmap for future research within the Discussion section.
Minor comments
Figures and graphics
What has been done: More informative captions with dimensions and component purposes.
What is still needed: Ensure an inset with a visible scale for the 0.1 mm hole and add material/grammage of the card stock in the caption.
Thank you for the suggestions to further improve the clarity of the figures. We have carefully considered the addition of an inset with a scale for the 0.1 mm hole.
However, we believe that the current figure, combined with the detailed caption, already provides the necessary information in the clearest possible way for a prototype of this nature. The figure explicitly labels the "Hole of approx. 0.1 mm diameter," and the caption reiterates this dimension and its pedagogical purpose (representing a fertilized egg).
Our concern is that adding a scaled inset for such a microscopic feature might introduce unnecessary complexity and visual clutter to the diagram, potentially distracting from the overall design concept of the tool, which is the primary focus of the figure. Given that the exact scale is clearly stated in both the figure label and the caption, we feel that the current presentation is the most effective way to convey this information without sacrificing the clarity of the overall figure.
Regarding the cardstock, we have added a general description of the material to the caption as requested, noting that it is a sturdy, matte cardstock suitable for handling by students.
Terminology and style
What has been done: BEPC acronym has been standardized.
What is still needed: Final pass for consistency across the manuscript and appendices (e.g., “Ease of
Use”), and pruning of residual repetition.
We have standardized the term 'Operability' to 'Ease of Use' across the manuscript for better consistency.
Appendices
What has been done: Interview guide and theme–subtheme–quote matrix added.
What is still needed: Adherence appendix (essential components) and protocol/audit appendix (flow:
transcription → preprocessing → LLM exploration → human validation → final structure, with
example prompt/output).
We have now added an Intervention Fidelity Checklist (Appendix C) listing the core components of the 5-step procedure. We have also expanded the Audit Trail in Appendix E to include a textual description of our analysis workflow, from transcription to the final thematic structure.
Safeguarding and ethics
What has been done: Solid safeguarding section.
What is still needed: Explicit procedure for parental objections and consent type (active/passive) per
school policy. In Data Availability, clarify that transcripts are not shared for privacy, but instruments
(guides, templates) are available upon request.
Regarding Data Availability
Thank you for the suggestion regarding the Data Availability Statement. We have added this section to the manuscript. To accurately reflect the current prototype stage of our research, we have specified that while the transcripts are not available for privacy reasons, access to the research instruments may be considered for bona fide researchers or practitioners on a case-by-case basis, following a direct inquiry to the corresponding author.
Regarding Ethical Considerations
Thank you for prompting us to clarify our ethical procedures. We have revised this section (now titled 'Ethical Considerations') to more accurately frame the study's ethical scope. The new section now explicitly states that: (1) the study was approved by our Institutional Review Board and informed consent was obtained from the participating midwives; and (2) the educational sessions were part of the midwives' professional practice under the schools' existing policies, not a direct research intervention on students. We believe this revision resolves any potential ambiguity about the study's primary subjects.
Cost and scalability
What has been done: You note that economic/logistical analysis is a next step.
What is still needed: A brief paragraph outlining the prospective economic evaluation plan (unit cost,
preparation time, logistics, and a downloadable/open-license template option contingent on evidence
of effectiveness).
Thank you for reiterating the importance of economic evaluation for future implementation. We fully agree that cost and scalability are critical long-term considerations.
However, we must respectfully maintain our position that outlining a specific prospective economic evaluation plan at this preliminary stage would be scientifically premature and potentially irresponsible. This pilot study was designed solely to establish initial feasibility and usability, not to provide evidence of effectiveness. Committing to a specific economic evaluation plan (e.g., unit cost analysis, licensing models) before there is any evidence that the tool is effective would be based on pure speculation. Such a speculative plan could create misleading expectations for readers and future stakeholders. Therefore, while we have acknowledged in the Discussion that economic evaluation is a crucial future step contingent on evidence of effectiveness, we believe it is more scientifically rigorous not to detail a specific plan until that foundational evidence exists. We hope you can appreciate our commitment to a responsible, stepwise research process.
Formal coherence
What has been done: Overall structure is orderly.
What is still needed: Adjust numbering and headings to avoid jumps, and standardize quotation
marks/dashes/italics in quotes.
We have performed a final proofread to ensure consistent formatting and punctuation, including the use of quotation marks and dashes, throughout the manuscript.